# Distributional Privacy for Data Sharing

**Zinan Lin**[*]
Carnegie Mellon University
Pittsburgh, PA 15213
zinanlin@microsoft.com

**Shuaiqi Wang**[*]
Carnegie Mellon University
Pittsburgh, PA 15213
shuaiqiw@andrew.cmu.edu

**Vyas Sekar**
Carnegie Mellon University
Pittsburgh, PA 15213
vsekar@andrew.cmu.edu

**Giulia Fanti**
Carnegie Mellon University
Pittsburgh, PA 15213
gfanti@andrew.cmu.edu

## Abstract

Data sharing between different parties has become an important engine powering modern research and development processes. An important class of privacy concerns in data sharing regards the underlying distribution of data. For example, the total traffic volume of data from a networking company reveals the scale of its business. Unfortunately, existing privacy frameworks do not adequately address this class of concerns. In this paper, we propose *distributional privacy*, a framework for analyzing and protecting these distributional privacy concerns in data sharing scenarios. Distributional privacy is applicable in multiple data sharing settings, including synthetic data release. Theoretically, we analyze the lower and upper bounds of privacy-distortion trade-offs. Practically, we propose data release mechanisms for protecting distributional privacy concerns, and demonstrate that they achieve better privacy-distortion trade-offs than alternative privacy mechanisms on real-world datasets.

## 1 Introduction

Data sharing between parties plays a critical role in modern research and development. Data holders typically either release processed data [18, 9, 5, 21] or train a generative model (e.g., GANs [8], diffusion models [11, 19, 20]) on their original data and share synthetic data with other parties. For example, network traces shared from customers to networking vendors enable vendors to debug and improve products [24, 1]. Medical data shared between hospitals [7, 21] enables them to develop new machine-learning-based diagnosis algorithms collaboratively [4]. More generally, data shared from researchers allow their research to be reproducible by others [15].

However, data sharing also raises privacy concerns. While the most well-studied privacy concerns relate to leaking user data, an important other class of concerns relates to the *underlying data distribution*. For example, a video analytics company that shares synthetic video session data may wish to hide the total or mean traffic volume, which could imply the company's total revenue [17]. A cloud provider that shares synthetic cluster performance traces may not want to reveal the proportions of different server types that the cloud provider owns, which are regarded as business secrets [15]. Note that this information (total/mean traffic volume, proportions of server types) cannot be inferred from any single record, but is inherent to the overall data distribution (or the aggregate dataset).

Existing privacy metrics and privacy-preserving data sharing algorithms do not adequately address these *distributional privacy concerns*, even in *synthetic data scenarios*. They either focus on protecting

---

[*]These authors contributed equally to this work.

NeurIPS 2022 Workshop on Synthetic Data for Empowering ML Research.

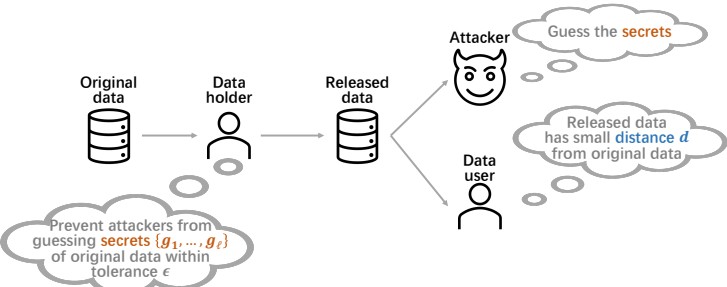

Figure 1: The related entities in a data sharing scenario. Data holder produces released data for the data user and wants to hide a set of *distributional secrets* $\{g_1, ..., g_l\}$ of the original data. The data user requires that the *distance* $d$ of released data from the original one is small. The attacker (could be the data user) also observes the released data, and wants to guess the *secrets* of the original data.

the privacy of individual records in a database (e.g., differential privacy [6], anonymization [18]), or are not suitable for synthetic data scenarios (e.g., attribute privacy [26], maximal leakage [12], privacy funnel [16]). See more discussions in "related work" below.

Motivated by these observations, we argue that a new privacy framework is needed for *defining, analyzing, and protecting distributional privacy concerns* in data sharing settings. In this paper, we take a first step towards the goal. At a high level, the proposed framework works as follows (detailed formulation in §2). A data holder first chooses one or more secrets, which are mathematically defined as functions of the data holder's data distribution. For example, a video analytics company might choose the mean daily observed traffic as a secret quantity. Then, the data holder obfuscates their data according to some *mechanism* and releases the output (Fig. 1). Our framework quantifies the privacy of this mechanism by analyzing the probability that an attacker can infer the data holder's true secret after observing the output. To capture the utility of released data, we define the distortion of a mechanism as the worst-case distance (where the distance metric can be chosen by the data holder or data user) between the original and released data distributions. Our goal is to design mechanisms that control tradeoffs between privacy and distortion.

**Related work.** Many existing techniques for privacy-preserving data release (both in the synthetic data setting and beyond) typically focus on protecting the privacy of individual records in a database. For instance, in a database of demographic information, these techniques might obfuscate whether a specific user's record is present in the database. Concretely, differential privacy [6] evaluates how much individual samples influence the final output of the algorithm. Similarly, popular privacy-preserving data sharing algorithms like sub-sampling [18], anonymization [18, 3], or more recently, differentially-private deep generative models [13, 7, 15, 24], focus on protecting sensitive information from individual samples. These techniques are not designed to protect distributional attributes at all (in fact, they are designed to preserve them).

Other frameworks like attribute privacy [26], maximal leakage [12], and privacy funnel [16] are designed to obfuscate sensitive random variables or properties of an aggregate dataset. They can be adapted to protect distributional privacy concerns. However, they are not necessarily well-suited to a data sharing setting. For instance, attribute privacy considers the setting of releasing a statistical query of the dataset (instead of a full synthetic dataset). Using it to release datasets would cause a poor privacy-distortion trade-off (see § 5). Maximal leakage considers the *worst-case* secrets in *discrete* space. But in our case, the secret is predefined by the data holder, and they could be in a continuous range (e.g., mean traffic). This seemingly small difference causes much of the analysis from [12] to not extend to our setting. Privacy funnel [16] evaluates the mutual information between a mechanism's output and input. This is ill-suited for data sharing, as a higher mutual information between the released and original datasets does not necessarily imply that they are close in distribution distance.

For readers interested in more technical details about these works, please refer to App. A.

**Contributions.** Our contributions are as follows.

- **Formulation:** We formalize distributional privacy concerns in data sharing applications with the notion of *distributional privacy* (§ 2). We propose new privacy and distortion metrics tailored to data sharing applications, including synthetic data sharing (§ 2).

- **Analysis (upper and lower bounds):** We derive fundamental limits (lower bounds) on the tradeoff between privacy and distortion for a specific choice of distortion distance metric. As a case study, we propose mechanisms that achieve order-optimal privacy-distortion tradeoffs when the secret is the mean of the distribution.
- **Practical evaluation:** We give preliminary empirical results showing how to use distributional privacy to release a real dataset, and how to evaluate the corresponding distributional privacy metric. We show that our proposed algorithms achieve better privacy-distortion trade-offs than baseline mechanisms that were designed to protect different privacy metrics (§ 5). Our proposed mechanisms are compatible with existing data synthesis tools (e.g., based on deep generative models [7, 15, 24, 13, 25]) and can be applied as an add-on to provide distributional privacy.

## 2 Distributional Privacy: Formulation

**Notation.** We denote random variables with uppercase letters, and their realizations with lowercase. For a random variable $X$, we denote its probability density function (PDF), or, in the case of discrete random variables, its probability mass function (PMF), as $f_X$, and its distribution measure as $\omega_X$. If a random variable $X$ is drawn from a parametric family (e.g., $X$ is Gaussian with specified mean and covariance); the parameters will be denoted with a subscript of $X$, i.e., the above notations become $X_\theta, f_{X_\theta}, \omega_{X_\theta}$ respectively for parameters $\theta \in \mathbb{R}^q$, where $q$ denotes the dimension of the parameters. In addition, we denote $f_{X|Y}$ as the conditional PDF/PMF of $X$ given another random variable $Y$.

**Original data.** Consider a data holder who possesses a dataset of $n$ samples $\mathcal{X} = \{x_1, \ldots, x_n\}$, where for each $i \in [n]$, $x_i \in \mathbb{R}^p$ is drawn i.i.d. from an underlying parametric distribution. That is, $x_i \sim \omega_{X_\theta}$, where $\theta \in \mathbb{R}^q$ is a realization of random $q$-dimensional parameter vector $\Theta$, and $\omega_\Theta$ is the probability measure for $\Theta$. In this paper, we assume that the data holder knows $\theta$; our results and mechanisms generalize it to the case when the data holder only possesses the dataset $\mathcal{X}$ (see § 4).

For example, suppose the original data samples come from a Gaussian distribution. We have $\theta = (\mu, \sigma)$, and $X_\theta \sim \mathcal{N}(\mu, \sigma)$. $\omega_\Theta$ (or $f_\Theta$) describes the prior distribution over $(\mu, \sigma)$. For example, if we know a priori that the mean of the Gaussian is drawn from a uniform distribution between 0 and 1, and $\sigma$ is always 1, we could have $f_\Theta(\mu, \sigma) = \mathbb{I}(\mu \in [0, 1]) \cdot \delta(\sigma)$, where $\mathbb{I}(\cdot)$ is the indicator function, and $\delta$ is the Dirac delta function. In practice, the underlying distribution can be much more complicated than a Gaussian.

**Distributional secrets to protect.** We assume the data holder wants to hide $\ell \in \mathbb{Z}_{>0}$ *secrets* from the original data distribution, expressed as a function $g(\theta) : \mathbb{R}^q \to \mathbb{R}^\ell$. In the Gaussian example $X_\theta \sim \mathcal{N}(\mu, \sigma)$, suppose the random variable $X_\theta$ represents the traffic volume experienced by an enterprise in a day. The data holder may wish to hide the mean traffic per day, in which case $g(\cdot)$ would be the mean of the distribution, i.e., $g(\mu, \sigma) = \mu$. In general, the secret can be any (vector-valued) function that can be deterministically computed from $\theta$. In this paper, we present general results for one-dimensional secrets (i.e., $\ell = 1$) and defer a discussion of higher-dimensional secrets to future work (see § 6).

**Data release mechanism.** The data holder releases data by passing the private parameter $\theta$ through a *data release mechanism* $\mathcal{M}_g$. That is, for a given $\theta$, the data holder first draws internal randomness $z \sim \omega_Z$, and then releases another distribution parameter $\theta' = \mathcal{M}_g(\theta, z)$, where $\mathcal{M}_g$ is a purely deterministic function, and $\omega_Z$ is a fixed distribution from which $z$ is sampled. We denote the random variable of the released data as $Y_{\theta'}$. Note that here we assume that both the input and output of $\mathcal{M}_g$ are distribution parameters. It is straightforward to generalize to the case when the input and/or output are datasets of samples (see § 4).

For example, in the Gaussian case discussed above, the data release mechanism can be $\mathcal{M}_g((\mu, \sigma), z) = (\mu + z, \sigma)$ where $z \sim \mathcal{N}(0, 1)$. I.e., this mechanism shifts the mean of the Gaussian by a random amount drawn from a standard Gaussian distribution and keeps the variance.

**Attacker strategy.** Based on the released parameter $\theta'$, the attacker outputs $\hat{g}(\theta')$ as a guess of the initial secret $g(\theta)$. $\hat{g}$ can be either random or deterministic. The attacker is assumed to know the data release mechanism $\mathcal{M}_g$ and output $\theta'$ but not the realization of the data holder's internal randomness $z$. For instance, in the running Gaussian example, an attacker may choose $\hat{g}(\mu', \sigma') = \mu'$. When the data holder releases a dataset of samples instead of the parameter $\theta'$, this formulation can be used to

upper bound attacker's performance as if the released dataset has unlimited amount of samples and therefore can estimate the distribution parameter correctly.

**Privacy metric.** The data holder cares about leaking its secrets. We define our privacy metric privacy $\Pi_\epsilon$ as the attacker's probability of guessing the secret(s) to within a tolerance $\epsilon$, worst-case over all attackers $\hat{g}$:

$$\Pi_\epsilon \triangleq \sup_{\hat{g}} \ \mathbb{P}\left(\hat{g}\left(\theta'\right) \in [g\left(\theta\right) - \epsilon, g\left(\theta\right) + \epsilon]\right) \ . \tag{1}$$

The probability is taken over the randomness of the original data distribution ($\theta \sim \omega_\Theta$), the data release mechanism ($z \sim \omega_Z$), and the attacker strategy ($\hat{g}$).

**Distortion metric.** The main goal of data sharing is to provide useful data; hence, we (and data holders and users) want to understand how much the released data distorts the original data. We define the *distortion* $\Delta$ of a mechanism as the worst-case distance between the original distribution and the released distribution:

$$\Delta \triangleq \sup_{\theta \in \text{Supp}(\omega_\Theta), \theta', z \in \text{Supp}(\omega_Z): \mathcal{M}_g(\theta, z) = \theta'} d\left(\omega_{X_\theta} \| \omega_{Y_{\theta'}}\right), \tag{2}$$

where $d$ is a general distance metric defined over distributions. The choice of the distance metric depends on the type of data and potentially depends on the applications that data holders and users care about. In this paper, we adopt Wasserstein-$\alpha$ distance as the starting point and all the results hold for any $\alpha \geq 1$. Wasserstein distance is often used for evaluating data quality (e.g., [24, 15]) and as the distance metric in neural network design (e.g., WGAN [2]). Note that the definition in Eq. (2) can be extended to data release mechanisms that take datasets as inputs and/or outputs.

**Objective.** To summarize, the data holder's objective is to choose a data release mechanism that minimizes privacy $\Pi_\epsilon$ subject to a constraint on distortion metric $\Delta$:

$$\min_{\mathcal{M}_g} \ \Pi_\epsilon \qquad \text{subject to } \Delta \leq T \tag{3}$$

An alternative formulation of minimizing $\Delta$ subject to a constraint on $\Pi_\epsilon$ is analyzed in App. F.

## 3 Lower Bounds: Privacy-Distortion Tradeoffs

In this section, we study lower bounds for Eq. (9); that is, given a budget for distortion $\Delta$, what are fundamental limits on achievable privacy metric $\Pi_\epsilon$?

We first present a lower bound that applies *regardless of the prior distribution of data* $\omega_\Theta$ and *regardless of the secret function* $g$. We assume that the secret is scalar (i.e., $\ell = 1$).

**Theorem 1** (Lower bound of privacy-distortion tradeoff)**.** *Let* $D\left(X_{\theta_1}, X_{\theta_2}\right) \triangleq \frac{1}{2} d\left(\omega_{X_{\theta_1}} \| \omega_{X_{\theta_2}}\right)$, *where* $d\left(\cdot \| \cdot\right)$ *denotes Wasserstein-$\alpha$ distance. Further, let* $R\left(X_{\theta_1}, X_{\theta_2}\right) \triangleq |g(\theta_1) - g(\theta_2)|$. *Let* $\gamma \triangleq \inf_{\theta_1, \theta_2 \in Supp(\omega_\Theta)} \frac{D\left(X_{\theta_1}, X_{\theta_2}\right)}{R\left(X_{\theta_1}, X_{\theta_2}\right)}$. *For any* $T > 0$, *when* $\Delta \leq T$, *we have* $\Pi_\epsilon \geq \lceil \frac{T}{2\gamma\epsilon}\rceil^{-1}$.

The proof is in App. B. Note that we have not made the quantity $\inf_{\theta_1, \theta_2 \in \text{Supp}(\omega_\Theta)} \frac{D\left(X_{\theta_1}, X_{\theta_2}\right)}{R\left(X_{\theta_1}, X_{\theta_2}\right)}$ in the lower bound explicit, as its closed-form expression depends on the secret function $g$. We have derived lower bounds for several scalar secret functions, including the mean, quantile, and standard deviation of continuous distributions, and the PMF of a discrete distribution evaluated at a specific point. Due to space constraints, we present only the result for the mean of a continuous distribution.

**Corollary 1** (Privacy lower bound, secret = mean of a continuous distribution)**.** *Consider the secret function* $g\left(\theta\right) = \int_x x f_{X_\theta}\left(x\right) dx$. *For any* $T > 0$, *when* $\Delta \leq T$, *we have* $\Pi_\epsilon \geq \lceil \frac{T}{\epsilon}\rceil^{-1}$.

The proof is in App. C. This lower bound is tight when $\epsilon$ divides $T$, as we show in the next section.

## 4 Upper bound: Data Release Mechanism Design

To satisfy distributional privacy, the data release mechanism (and corresponding privacy-distortion tradeoffs) depend on the secret function and potentially on the original data distribution. Our

high-level vision is to build a library of mechanisms for different common secret functions and families of parametric distributions; the data holder would then choose the setting that most closely matches their own data. We have preliminary mechanism designs and analysis for a number of secret functions (e.g., mean, $\alpha$-quantile, PMF value) and data distributions (e.g., Gaussian, exponential, Pareto, geometric). Due to space constraints, we present here results when the secret function is the mean of a one-dimensional continuous distribution that can be parameterized with a location parameter. That is, the distribution parameter vector $\theta$ can be written as $(u, v)$, where $u \in \mathbb{R}$, $v \in \mathbb{R}^{q-1}$, and for any $u \neq u'$, $f_{X_{u,v}}(x) = f_{X_{u',v}}(x - u' + u)$. The prior over distribution parameters is $f_{U,V}(a, b) = f_U(a) \cdot f_V(b)$, where $f_U(a) = \frac{1}{\overline{u} - \underline{u}} \mathbb{I}(a \in [\underline{u}, \overline{u}])$. In other words, the prior distribution of the location parameter is uniform and independent of other factors. Examples include the Gaussian, Laplace, and uniform distributions, as well as shifted distributions (e.g., shifted exponential, shifted log-logistic).

---

**Algorithm 1:** Shifting data release mechanism

---

**input** : $\theta = (u, v)$ where $u$ is the mean of $X_\theta = X_{u,v}$, and $v \in \mathbb{R}^{q-1}$ controls other properties of the distribution, lower bound $\underline{u}$ of the mean value $u$, quantization interval $s$.

1   $u' \leftarrow \underline{u} + \left( \lfloor \frac{u - \underline{u}}{s} \rfloor + 0.5 \right) \cdot s$ ;

2   $v' \leftarrow v$;

3   $\mathcal{M}_g((u, v), z) \leftarrow (u', v')$.

---

**Shifting data release mechanism.** The data release mechanism we design is shown in Alg. 1. $s$ is a hyperparameter of the mechanism that divides $(\overline{u} - \underline{u})$, i.e., there exists $N \in \mathbb{N}$ such that $s = \frac{\overline{u} - \underline{u}}{N}$. Fig. 2 shows an example when the original data distribution is Gaussian, i.e., $X_\theta \sim \mathcal{N}(u, v)$, and $u \in [\underline{u}, \overline{u}]$. Our mechanism "quantizes" the range of possible mean values into segments of length $s$. It then shifts the mean of private distribution $f_{X_{u,v}}$ to the midpoint of its corresponding segment, and releases the resulting distribution $f_{Y_{u',v'}}$. This simple deterministic mechanism is able to achieve the optimal privacy-distortion tradeoff in some cases, as shown below.

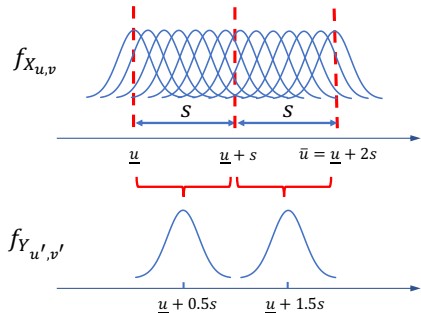

Figure 2: Illustration of the shifting data release mechanism.

**Theorem 2** (Optimality of Alg. 1 ). *When $2\epsilon$ divides $s$, the data release mechanism in Alg. 1 achieves the lower bound in Thm. 1.*

The proof is in App. D. Let $\hat{g}^*$ be the attack strategy that $\hat{g}^*(u', v') = u'$, i.e., the attack guesses the mean of private distribution as the release mean. We also show in the proof that under the data release mechanism Alg. 1, $\hat{g}^*$ is the optimal attacker strategy (i.e., $\hat{g}^* \in \arg\sup_{\hat{g}} \mathbb{P}(\hat{g}(\theta') \in [g(\theta) - \epsilon, g(\theta) + \epsilon])$).

**Extending the data release mechanism for handling dataset input/output.** Suppose the data holder possesses a dataset of i.i.d. samples $\mathcal{X} = \{x_1, \ldots, x_n\}$, where $x_i \sim \omega_{X_{u,v}}$, $i \in [n]$. The data release mechanism can be trivially extended to the dataset by shifting individual samples by the same amount as the distribution shift in Alg. 1:

$$\mathcal{M}_g(\mathcal{X}, z) = \{m(x_i)\}_{i=1}^n, \qquad \text{where } m(x) = x_j - u + \underline{u} + \left( \lfloor \frac{u - \underline{u}}{s} \rfloor + 0.5 \right) \cdot s. \qquad (4)$$

As before, $s$ is a hyperparameter of the mechanism that divides $(\overline{u} - \underline{u})$. In practice, since we do not know the true mean $u$, we estimate it by $\hat{u} = \frac{1}{n} \sum_{i \in [n]} x_i$. Note that this mechanism applies to samples. Therefore, it can be applied either to the original data, or as an add-on to existing data sharing tools [7, 15, 24, 13, 25]. For example, it can be used to modify synthetically-generated samples after they are generated, or to modify the training dataset for a generative model, or to directly modify the original data for releasing.

# 5 Experiments

In this section, we experimentally compare our mechanism for achieving *distributional privacy* with privacy mechanisms that were designed to satisfy other privacy notions (e.g., differential privacy). We demonstrate that prior mechanisms incur poor privacy-distortion tradeoffs under our metrics.

**Dataset.** We use Wikipedia Web Traffic Dataset [9] for the experiment. This dataset contains the daily page views of 145,063 Wikipedia web pages from Jul. 1st, 2015 to Dec. 31st, 2016. To preprocess it for our experiments, we remove the web pages with empty page view record on any day (117,277 left), and compute the mean page views across all dates for each web page. The goal is to release the page views (i.e., a 117,277-dimensional vector) while protecting the mean of the distribution (which reveals the business scales of the company § 1).

**Baselines.** We compare our mechanisms discussed in § 4 with two popular mechanisms proposed in prior work (§ 1): differentially-private density estimation [22] (shortened to DP) and attribute-private Gaussian mechanism [26] (shortened to AP).

For a dataset of samples $\mathcal{X} = \{x_1, ..., x_n\}$, DP works by: (1) Dividing the space into $m$ bins: $B_1, ..., B_m$. (2) Computing the histogram $C_i = \sum_{j=1}^{n} \mathbb{I}(x_j \in B_i)$. (3) Adding noise to the histograms $D_i = \max\{0, C_i + \text{Laplace}(0, \epsilon)\}$, where $\text{Laplace}(0, \epsilon^2)$ means a random noise from Laplace distribution with mean 0 and variance $\epsilon^2$. (4) Normalizing the histogram $p_i = \frac{D_i}{\sum_{j=1}^{m} D_j}$. We can then draw $y_i$ according to the histogram and release $\mathcal{Y} = \{y_1, ..., y_n\}$ with differential privacy guarantee. AP works by releasing $\mathcal{Y} = \{x_i + \mathcal{N}(0, \epsilon^2)\}_{i=1}^{n}$.

**Metrics.** Our privacy and distortion metrics depend on the prior distribution of the original data $\theta \sim \omega_\Theta$ (though the mechanism does not). In practice (and also in this experiment), the data holder only has one dataset. Therefore, we cannot empirically evaluate the proposed privacy and distortion metrics, and resort to surrogate metrics to bound our true privacy and distortion.

*Surrogate privacy metric.* For an original dataset $\mathcal{X} = \{x_1, ..., x_n\}$ and the released dataset $\mathcal{Y} = \{y_1, ..., y_n\}$, we define the surrogate privacy metric $\tilde{\Pi}_\epsilon$ as the error of an attacker who guesses the mean of the released dataset as the secret: $\tilde{\Pi}_\epsilon \triangleq -\left| \frac{1}{n} \sum_{i=1}^{n} x_i - \frac{1}{n} \sum_{i=1}^{n} y_i \right|$. Note that a minus sign is added here so that a smaller value indicates stronger privacy, as in privacy metric (1). This simple attacker strategy is in fact a good proxy for evaluating the privacy $\Pi_\epsilon$ due to the following facts. (1) For our mechanism Alg. 1, when the prior distribution is uniform, this strategy is actually optimal (see § 4), so there is a direct mapping between $\tilde{\Pi}_\epsilon$ and $\Pi_\epsilon$. (2) For AP, this strategy gives an unbiased estimator of the secret. (3) For DP, this mechanism may not be an unbiased estimator of the secret, but it gives an *upper bound* on the attacker's error.

*Surrogate distortion metric.* We define our surrogate distortion metric as the Wasserstein-1 distance between the two datasets: $\tilde{\Delta} \triangleq d_{\text{Wasserstein-1}}(p_\mathcal{X} \| p_\mathcal{Y})$ where $p_D$ denotes the empirical distribution of a dataset $D$. This metric evaluates how much the mechanism distorts the dataset.

In fact, we can deduce a theoretical lower bound for the surrogate privacy and distortion metrics using similar techniques as Corollary 2 (App. E).

**Results.** We enumerate the hyper-parameters of each method (bin size and $\epsilon$ for DP, $\epsilon$ for AP, and $s$ for ours). For each method and each hyper-parameter, we compute their surrogate privacy and distortion metrics. The results are shown in Fig. 3 (bottom left is best); each data point represents one realization of mechanism $\mathcal{M}_g$ under a distinct hyperparameter setting. AP directly adds Gaussian noise to each sample. This process does not change the mean of the distribution on expectation. Therefore, Figure 3 shows that AP has a bad privacy-distortion tradeoff. DP quantizes (bins) the samples before adding noise. Quantization has a better property in terms of protecting the mean of the distribution, and therefore we see that DP has a better privacy-distortion

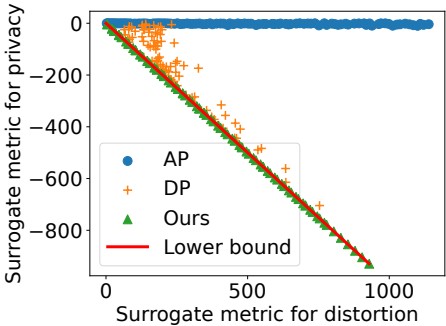

Figure 3: Privacy (lower is better) and distortion (lower is better) of AP, DP and ours on Wikipedia Web Traffic Dataset. Each point represents one instance of data release mechanism with one hyper-parameter.

tradeoff than AP. Our mechanism (Eq. (4)) always achieves the best privacy-distortion tradeoff; indeed, it can be shown to be optimal even for these surrogate metrics (proofs in App. E).

# 6 Discussions and Future Work

This paper gives an initial framework for defining, analyzing, and protecting distributional privacy. Many open questions remain, including how to measure privacy empirically for arbitrary secrets and data distributions, how to analyze the privacy-distortion tradeoff (and design mechanisms) for more secret types, distribution classes (including parametric families without an explicit location parameter), multiple secrets, higher dimension of data, and other distance metrics for distortion. While our framework is not intended for protecting sample level privacy (see § 1), it would be interesting to study whether our mechanisms (§ 4) are able to defend against sample-level attacks (e.g., membership inference attacks [10]) as a side benefit.

## Acknowledgement

This research was sponsored in part by the National Science Foundation RINGS award 2148359 and by the Air Force Office of Scientific Research under award number FA9550-21-1-0090. This work was also supported in part by faculty research awards from JP Morgan Chase, Intel, and the Sloan Foundation, as well as gift grants from Cisco and Siemens AG. This material is based upon work supported by the U.S. Army Research Office and the U.S. Army Futures Command under Contract No. W911NF20D0002. The content of the information does not necessarily reflect the position or the policy of the government and no official endorsement should be inferred.

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

# Appendix

## A Related Work

In this section, we discuss why existing privacy frameworks and mechanisms are unsuitable for this distributional privacy concern.

**Differential privacy (DP)** [6] is one of the most popular privacy notions. A randomized mechanism $\mathcal{M}$ is $(\epsilon, \delta)$-differentially-private if for any neighboring datasets $D_0$ and $D_1$ (i.e., $D_0$ and $D_1$ differ one sample), and any set $S \subseteq range\,(\mathcal{M})$, we have

$$\mathbb{P}\,(\mathcal{M}\,(D_0) \in S) \leq e^{\epsilon} \cdot \mathbb{P}\,(\mathcal{M}\,(D_1) \in S) + \delta \ .$$

In our data sharing scenarios, we could apply DP framework by treating $\mathcal{M}$ as the data release mechanism that reads the original dataset and outputs the released dataset. However, the privacy concerns of DP and our suggested framework are completely different: we aim to hide functions of *a distribution*, while DP aims to hide whether *any given sample* contributed to the shared data. Indeed, we will show in § 5 that DP is not effective in hiding distributional secrets.

**Maximal leakage** [12] is a framework for quantifying the leakage of sensitive information. We denote $X$ as the random variable of the data that many contain sensitive information, and $Y$ as the random variable of the information that is processed from $X$ and is accessible by the attacker. With the observation of $Y$, the attacker's goal is to guess a secret function of $X$ denoted by $U$, and the guess is denoted by $\hat{U}$. Based on this setup, the Markov chain $U - X - Y - \hat{U}$ holds. Maximal leakage from $X$ to $Y$ is defined as

$$\mathcal{L}\,(X \to Y) = \sup_{U - X - Y - \hat{U}} \log \frac{\mathbb{P}\left(U = \hat{U}\right)}{\max_u P_U(u)} \tag{5}$$

where the sup is taking over $U$ (i.e., considering the worst-case secret) and $\hat{U}$ (i.e., considering the strongest attacker). Intuitively, Eq. (5) evaluates the ratio (in nats) of the probabilities of guessing the secret correctly with and without observing $Y$.

To apply maximal leakage in data sharing scenario, we may regard $X$ as the original dataset, $Y$ as the released dataset, and $U$ as the secret (e.g., the fraction of a specific server type). However, this formulation is still unsuitable for the following reasons. (1) Maximal leakage only considers discrete $U$ and $\hat{U}$ under finite alphabet. Note that it is a critical assumption for making sure that $\mathbb{P}\left(U = \hat{U}\right)$ in the definition (Eq. (5)) is nontrivial. However, in our problem, secrets are usually in continuous domains (e.g., the secrets in the motivating examples in § 1). (2) Maximal leakage assumes that the secret to protect $U$ is unknown a priori and therefore considers the worst-case leakage among all possible secrets. However, in our problem, data holders know what secret they want to protect.

**Attribute privacy** [26] considers a similar privacy concerns as us: it tries to protect a function of a sensitive column in the dataset (named *dataset attribute privacy*) or a sensitive parameter of the underlying distribution from which the data is sampled (named *distribution attribute privacy*). Attribute privacy applies *Pufferfish privacy framework* [14] to describe these two privacy concerns. On the high level, an algorithm is said to be of higher dataset/distribution attribute privacy if for any two different ranges of secret (e.g., fraction of the server type is in $[0.1, 0.2)$ or $[0.2, 0.3)$), the distributions of the algorithm output do not differ too much. For the more accurate definitions, please refer to [26].

Although the privacy concerns are highly related, attribute privacy focuses on algorithms that output *a statistical query of the dataset* (e.g., the number of servers) instead of the entire dataset (e.g., cluster trace dataset). We could still apply the framework to analyze such data sharing algorithms, but due to the high dimensionality of the dataset, attribute privacy requires to add too much noise which makes the generated dataset of high distortion. Indeed, in § 5 we will see that attribute privacy mechanisms give poor privacy-distortion trade-offs in data sharing applications.

**Privacy funnel** [16] is another popular privacy framework. We denote $X$ as the random variable of the data that many contain sensitive information $U$, and $Y$ as the random variable of the information that is processed from $X$ and is accessible by the attacker. The privacy funnel framework use the

mutual information $I(U;Y)$ to evaluate the privacy leakage, and use the mutual information $I(X;Y)$, to evaluate how well $Y$ preserves the utility of data. To find a good privacy-preserving data processing strategy $P_{Y|X}$, privacy funnel solves the optimization problem

$$\min_{P_{Y|X}:I(X;Y)\geq R} I(S;Y) \; ,$$

where $R$ is a desired threshold on the utility of $Y$.

To apply it in data sharing problems, we could regard $X$ as the original data, $Y$ as the released data, and $U$ as the secret data holder wants to protect (e.g., the fraction of a specific server type). However, mutual information is not a good metrics for either privacy or utility. On the privacy front, prior work has shown that we could make $I(S;Y)$ smaller while allowing the attacker to guess $S$ correctly from $Y$ with higher probability (see Example 1 in [12]). On the utility front, higher mutual information $I(X;Y)$ does not mean that the released data $Y$ is a useful representation of $X$. For example, for could have $Y$ as an arbitrary one-to-one transformation of $X$. In that case, $I(X;Y)$ obtains the maximum but the structure of data could be completely destroyed. In addition, privacy funnel [16] only considers $X$ and $Y$ in discrete domains, which is not the case for our setting.

**Sub-sampling** is used in prior data release practice for reducing the leakage of sensitive information [18]. It works by sub-sampling the original datasets before doing follow-up processing or release. The intuition is that by reducing the number of involved samples, less information can be leaked. However, sub-sampling does not change the statistical properties of the distribution. Distributional information can still be inferred from the released data (possibly with higher estimation variance, though).

**Anonymization**, such as removing certain attributes (e.g., name of the patients in medical data, name of jobs in cluster dataset) [18], is widely used in the release process of public datasets (e.g., [23]). However, this process does not change the distribution of other attributes, which may contain the private information that data holder wants to hide [15].

*Summary.* Based on the above observation, it is important to establish a new privacy framework for distributional privacy in data sharing scenarios, which we discuss in the next section.

## B  Proof of Thm. 1

*Proof.* For any $\theta'$, we have

$$
\begin{aligned}
T &\geq \Delta \\
&\geq \sup_{\theta\in\text{Supp}(\omega_\Theta),z\in\text{Supp}(\omega_Z):\mathcal{M}_g(\theta,z)=\theta'} d\left(\omega_{X_\theta}\|\omega_{Y_{\theta'}}\right) \\
&\geq \sup_{\theta_i\in\text{Supp}(\omega_\Theta),z_i:\mathcal{M}_g(\theta_i,z_i)=\theta'} D\left(X_{\theta_1},X_{\theta_2}\right) \qquad (6) \\
&\geq \gamma\cdot \sup_{\theta_i\in\text{Supp}(\omega_\Theta),z_i:\mathcal{M}_g(\theta_i,z_i)=\theta'} R\left(X_{\theta_1},X_{\theta_2}\right)
\end{aligned}
$$

where Eq. (6) comes from triangle inequality.

Let

$$
\begin{aligned}
L_{\theta'} &\triangleq \inf_{\theta\in\text{Supp}(\omega_\Theta),z:\mathcal{M}_g(\theta,z)=\theta'} g\left(\theta\right) \; , \\
R_{\theta'} &\triangleq \sup_{\theta\in\text{Supp}(\omega_\Theta),z:\mathcal{M}_g(\theta,z)=\theta'} g\left(\theta\right) \; .
\end{aligned}
$$

From the above result, we know that $R_{\theta'} - L_{\theta'} \leq \frac{T}{\gamma}$. We can define a sequence of attackers such that $\hat{g}_i\left(\theta'\right) = L_{\theta'} + (i+0.5)\cdot 2\epsilon$ for $i\in\left\{0,1,\ldots,\lceil\frac{T}{2\gamma\epsilon}\rceil-1\right\}$ (Fig. 4). We have

$$\sum_i \mathbb{P}\left(\hat{g}_i\left(\theta'\right)\in[g\left(\theta\right)-\epsilon,g\left(\theta\right)+\epsilon]\Big|\theta'\right) \geq 1,$$

and therefore,

$$\max_i \mathbb{P}\left(\hat{g}_i\left(\theta'\right)\in[g\left(\theta\right)-\epsilon,g\left(\theta\right)+\epsilon]\Big|\theta'\right) \geq \lceil\frac{T}{2\gamma\epsilon}\rceil^{-1}, \qquad (7)$$

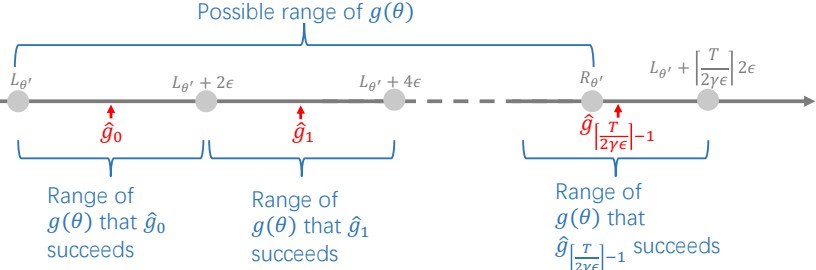

Figure 4: The construction of attackers for proof of Thm. 1. The $2\epsilon$ ranges of $\hat{g}_0, ..., \hat{g}_{\lceil \frac{T}{2\gamma\epsilon} \rceil - 1}$ jointly cover the entire range of possible secret $[L_{\theta'}, R_{\theta'}]$. Therefore, there exists one attacker whose probability of guessing the secret correctly within $\epsilon$ is $\geq \lceil \frac{T}{2\gamma\epsilon} \rceil^{-1}$ (Eq. (7)).

which implies that

$$\sup_{\hat{g}} \mathbb{P}\left( \hat{g}(\theta') \in [g(\theta) - \epsilon, g(\theta) + \epsilon] \, \middle| \, \theta' \right) \geq \lceil \frac{T}{2\gamma\epsilon} \rceil^{-1}.$$

Therefore, we have

$$\Pi_\epsilon = \sup_{\hat{g}} \mathbb{P}\left( \hat{g}(\theta') \in [g(\theta) - \epsilon, g(\theta) + \epsilon] \right)$$

$$= \sup_{\hat{g}} \mathbb{E}\left( \mathbb{P}\left( \hat{g}(\theta') \in [g(\theta) - \epsilon, g(\theta) + \epsilon] \, \middle| \, \theta' \right) \right)$$

$$= \mathbb{E}\left( \sup_{\hat{g}} \mathbb{P}\left( \hat{g}(\theta') \in [g(\theta) - \epsilon, g(\theta) + \epsilon] \, \middle| \, \theta' \right) \right)$$

$$\geq \lceil \frac{T}{2\gamma\epsilon} \rceil^{-1}.$$

$\square$

## C  Proof of Corollary 1

*Proof.* For any $X_{\theta_1}, X_{\theta_2}$, we have

$$D\left( X_{\theta_1}, X_{\theta_2} \right) = \frac{1}{2} d_{\text{Wasserstein-}\alpha}\left( \omega_{X_{\theta_1}} \| \omega_{X_{\theta_2}} \right)$$

$$\geq \frac{1}{2} |g(\theta_1) - g(\theta_2)| \tag{8}$$

$$= \frac{1}{2} R\left( X_{\theta_1}, X_{\theta_2} \right).$$

where Eq. (8) comes from Jensen inequality. Therefore, we have $\gamma = \inf_{\theta_1, \theta_2 \in \text{Supp}(\omega_\Theta)} \frac{D\left( X_{\theta_1}, X_{\theta_2} \right)}{R\left( X_{\theta_1}, X_{\theta_2} \right)} \geq \frac{1}{2}$. The result then follows from Thm. 1. $\square$

# D Proof of Thm. 2

*Proof.* For any released parameter $\theta' = (u', v')$, there exists $i \in \{0, ..., N-1\}$ such that $u' = \underline{u} + (i + 0.5) \cdot s$. We have

$$\sup_{\hat{g}} \mathbb{P}\left(\hat{g}\left(\theta'\right) \in [g\left(\theta\right) - \epsilon, g\left(\theta\right) + \epsilon] \,\middle|\, \theta'\right)$$

$$= \sup_{\hat{g}} \int_{\underline{u}+i\cdot s}^{\underline{u}+(i+1)\cdot s} f_{U|U'}\left(u|u'\right) \cdot \int_{u-\epsilon}^{u+\epsilon} f_{\hat{g}(u',v')}\left(h\right) \, \mathrm{d}h \, \mathrm{d}u$$

$$= \sup_{\hat{g}} \int_{\underline{u}+i\cdot s-\epsilon}^{\underline{u}+(i+1)\cdot s+\epsilon} f_{\hat{g}(u',v')}(h) \cdot \int_{\hat{g}\left(f_{Y_{u',v'}}\right)-\epsilon}^{\hat{g}\left(f_{Y_{u',v'}}\right)+\epsilon} f_{U|U'}\left(u|u'\right) \, \mathrm{d}u \, \mathrm{d}h$$

$$\leq \sup_{\hat{g}} \int_{\underline{u}+i\cdot s-\epsilon}^{\underline{u}+(i+1)\cdot s+\epsilon} \frac{2\epsilon}{s} \cdot f_{\hat{g}(u',v')}(h) \, \mathrm{d}h$$

$$\leq \frac{2\epsilon}{s}.$$

Therefore, we have

$$\Pi_\epsilon = \sup_{\hat{g}} \mathbb{P}\left(\hat{g}\left(\theta'\right) \in [g\left(\theta\right) - \epsilon, g\left(\theta\right) + \epsilon]\right)$$

$$= \sup_{\hat{g}} \mathbb{E}\left(\mathbb{P}\left(\hat{g}\left(\theta'\right) \in [g\left(\theta\right) - \epsilon, g\left(\theta\right) + \epsilon] \,\middle|\, \theta'\right)\right)$$

$$= \mathbb{E}\left(\sup_{\hat{g}} \mathbb{P}\left(\hat{g}\left(\theta'\right) \in [g\left(\theta\right) - \epsilon, g\left(\theta\right) + \epsilon] \,\middle|\, \theta'\right)\right)$$

$$\leq \frac{2\epsilon}{s}.$$

Let $\hat{g}^*$ be the attack strategy that $\hat{g}^*\left(u', v'\right) = u'$. When $\hat{g} = \hat{g}^*$, the equality achieves, and thus $\hat{g}^* \in \arg\sup_{\hat{g}} \mathbb{P}\left(\hat{g}\left(\theta'\right) \in [g\left(\theta\right) - \epsilon, g\left(\theta\right) + \epsilon]\right)$.

For the distortion, we can easily get that $\Delta = \frac{s}{2}$. From Corollary 1, we know that $\Pi_\epsilon \geq \lceil \frac{s}{2\epsilon} \rceil^{-1}$. Therefore, when $2\epsilon$ divides $s$, the data release mechanism in Alg. 1 is optimal. $\qquad\square$

# E Proofs for the Surrogate Metrics

For any $p_\mathcal{Y}$, we have $\tilde{\Delta} = d_{\text{Wasserstein-1}}\left(p_\mathcal{X} \| p_\mathcal{Y}\right) \geq \left| \frac{1}{n} \sum_{i=1}^n x_i - \frac{1}{n} \sum_{i=1}^n y_i \right| = -\tilde{\Pi}_\epsilon$.

For $p_\mathcal{Y}$ is released from our mechanism Eq. (4), we have $\tilde{\Delta} = d_{\text{Wasserstein-1}}\left(p_\mathcal{X} \| p_\mathcal{Y}\right) = \left| \frac{1}{n} \sum_{i=1}^n x_i - \frac{1}{n} \sum_{i=1}^n y_i \right| = -\tilde{\Pi}_\epsilon$.

# F Analysis of the Alternative Formulation

In this section, we present the alternative formulation of minimizing distortion $\Delta$ subject to a constraint on privacy metric $\Pi_\epsilon$:

$$\min_{\mathcal{M}_g} \Delta \qquad \text{subject to } \Pi_\epsilon \leq T \tag{9}$$

**Theorem 3** (Lower bound of privacy-distortion tradeoff). *Let* $D\left(X_{\theta_1}, X_{\theta_2}\right) \triangleq \frac{1}{2} d\left(\omega_{X_{\theta_1}} \| \omega_{X_{\theta_2}}\right)$, *where* $d\left(\cdot \| \cdot\right)$ *denotes Wasserstein-$\alpha$ distance. Further, let* $R\left(X_{\theta_1}, X_{\theta_2}\right) \triangleq |g(\theta_1) - g(\theta_2)|$. *Let* $\gamma \triangleq \inf_{\theta_1, \theta_2 \in Supp(\omega_\Theta)} \frac{D\left(X_{\theta_1}, X_{\theta_2}\right)}{R\left(X_{\theta_1}, X_{\theta_2}\right)}$. *For any* $T \in [0, 1]$, *when* $\Pi_\epsilon \leq T$, *we have* $\Delta > \left(\lceil \frac{1}{T} \rceil - 1\right) \cdot 2\gamma\epsilon$.

The proof is in App. G.

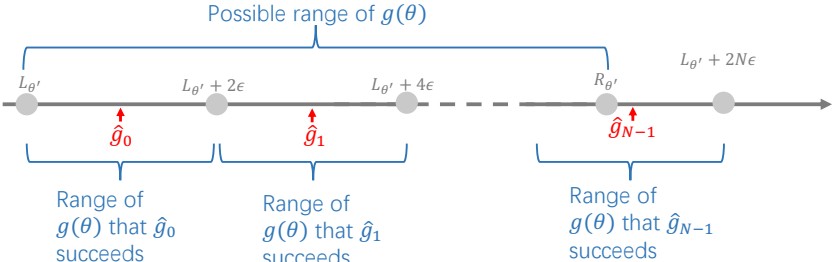

Figure 5: The construction of attackers for proof of Thm. 3. The $2\epsilon$ ranges of $\hat{g}_0, ..., \hat{g}_{N-1}$ jointly cover the entire range of possible secret $[L_{\theta'}, R_{\theta'}]$. Therefore, there exists one attacker whose probability of guessing the secret correctly within $\epsilon$ is $\geq \lceil \frac{T}{2\gamma\epsilon} \rceil^{-1}$ (Eq. (7)).

**Corollary 2** (Privacy lower bound, secret = mean of a continuous distribution). *Consider the secret function* $g(\theta) = \int_x x f_{X_\theta}(x) \, dx$. *For any* $T \in [0, 1)$, *when* $\Pi_\epsilon \leq T$, *we have* $\Delta > \left( \lceil \frac{1}{T} \rceil - 1 \right) \cdot \epsilon$.

The proof is similar to App. C except that the last step applies Thm. 3 instead of Thm. 1.

**Theorem 4** (Optimality of Alg. 1). *The data release mechanism in Alg. 1 asymptotically achieves the lower bound in Thm. 3 when* $\epsilon \to 0$.

The proof is in App. H. Let $\hat{g}^*$ be the attack strategy that $\hat{g}^*(u', v') = u'$, i.e., the attack guesses the mean of private distribution as the release mean. We also show in the proof that under the data release mechanism Alg. 1, $\hat{g}^*$ is the optimal attacker strategy (i.e., $\hat{g}^* \in \arg\sup_{\hat{g}} \mathbb{P}(\hat{g}(\theta') \in [g(\theta) - \epsilon, g(\theta) + \epsilon])$).

## G  Proof of Thm. 3

*Proof.*

$$T \geq \Pi_\epsilon$$
$$= \sup_{\hat{g}} \mathbb{P}\left(\hat{g}(\theta') \in [g(\theta) - \epsilon, g(\theta) + \epsilon]\right)$$
$$= \sup_{\hat{g}} \mathbb{E}\left(\mathbb{P}\left(\hat{g}(\theta') \in [g(\theta) - \epsilon, g(\theta) + \epsilon] \Big| \theta'\right)\right)$$
$$= \mathbb{E}\left(\sup_{\hat{g}} \mathbb{P}\left(\hat{g}(\theta') \in [g(\theta) - \epsilon, g(\theta) + \epsilon] \Big| \theta'\right)\right)$$

Therefore, there exists $\theta'$ s.t. $\sup_{\hat{g}} \mathbb{P}\left(\hat{g}(\theta') \in [g(\theta) - \epsilon, g(\theta) + \epsilon] \Big| \theta'\right) \leq T$.

Let

$$L_{\theta'} \triangleq \inf_{\theta \in \text{Supp}(\omega_\Theta), z: \mathcal{M}_g(\theta, z) = \theta'} g(\theta) \ ,$$

$$R_{\theta'} \triangleq \sup_{\theta \in \text{Supp}(\omega_\Theta), z: \mathcal{M}_g(\theta, z) = \theta'} g(\theta) \ .$$

We can define a sequence of attackers such that $\hat{g}_i(\theta') = L_{\theta'} + (i + 0.5) \cdot 2\epsilon$ for $i \in \{0, 1, \ldots, N - 1\}$ and $L_{\theta'} + 2N\epsilon \geq R_{\theta'} > L_{\theta'} + 2(N - 1)\epsilon$ (Fig. 5). From the above, we have

$$T \cdot N \geq \sum_i \mathbb{P}\left(\hat{g}_i(\theta') \in [g(\theta) - \epsilon, g(\theta) + \epsilon] \Big| \theta'\right) \geq 1,$$

Therefore, we have $N \geq \lceil \frac{1}{T} \rceil$, and $R_{\theta'} - L_{\theta'} > \left( \lceil \frac{1}{T} \rceil - 1 \right) \cdot 2\epsilon$. Then we have

$$
\Delta \geq \sup_{\theta \in \mathrm{Supp}(\omega_\Theta), z \in \mathrm{Supp}(\omega_Z): \mathcal{M}_g(\theta, z) = \theta'} d\left( \omega_{X_\theta} \| \omega_{Y_{\theta'}} \right)
$$

$$
\geq \sup_{\theta_i \in \mathrm{Supp}(\omega_\Theta), z_i: \mathcal{M}_g(\theta_i, z_i) = \theta'} D\left( X_{\theta_1}, X_{\theta_2} \right) \tag{10}
$$

$$
> \left( \lceil \frac{1}{T} \rceil - 1 \right) \cdot 2\gamma\epsilon. \tag{11}
$$

where Eq. (11) utilizes $R_{\theta'} - L_{\theta'} > \left( \lceil \frac{1}{T} \rceil - 1 \right) \cdot 2\epsilon$ and the definition of $\gamma$, and Eq. (10) comes from triangle inequality. $\qquad\square$

## H   Proof of Thm. 4

*Proof.* For any released parameter $\theta' = (u', v')$, there exists $i \in \{0, ..., N-1\}$ such that $u' = \underline{u} + (i + 0.5) \cdot s$. We have

$$
\sup_{\hat{g}} \mathbb{P}\left( \hat{g}(\theta') \in [g(\theta) - \epsilon, g(\theta) + \epsilon] \,\big|\, \theta' \right)
$$

$$
= \sup_{\hat{g}} \int_{\underline{u}+i\cdot s}^{\underline{u}+(i+1)\cdot s} f_{U|U'}(u|u') \cdot \int_{u-\epsilon}^{u+\epsilon} f_{\hat{g}(u', v')}(h) \ \mathrm{d}h \ \mathrm{d}u
$$

$$
= \sup_{\hat{g}} \int_{\underline{u}+i\cdot s-\epsilon}^{\underline{u}+(i+1)\cdot s+\epsilon} f_{\hat{g}(u', v')}(h) \cdot \int_{\hat{g}\left(f_{Y_{u', v'}}\right)-\epsilon}^{\hat{g}\left(f_{Y_{u', v'}}\right)+\epsilon} f_{U|U'}(u|u') \ \mathrm{d}u \ \mathrm{d}h
$$

$$
\leq \sup_{\hat{g}} \int_{\underline{u}+i\cdot s-\epsilon}^{\underline{u}+(i+1)\cdot s+\epsilon} \frac{2\epsilon}{s} \cdot f_{\hat{g}(u', v')}(h) \ \mathrm{d}h
$$

$$
\leq \frac{2\epsilon}{s}.
$$

Therefore, we have

$$
\Pi_\epsilon = \sup_{\hat{g}} \mathbb{P}\left( \hat{g}(\theta') \in [g(\theta) - \epsilon, g(\theta) + \epsilon] \right)
$$

$$
= \sup_{\hat{g}} \mathbb{E}\left( \mathbb{P}\left( \hat{g}(\theta') \in [g(\theta) - \epsilon, g(\theta) + \epsilon] \,\Big|\, \theta' \right) \right)
$$

$$
= \mathbb{E}\left( \sup_{\hat{g}} \mathbb{P}\left( \hat{g}(\theta') \in [g(\theta) - \epsilon, g(\theta) + \epsilon] \,\Big|\, \theta' \right) \right)
$$

$$
\leq \frac{2\epsilon}{s}.
$$

Let $\hat{g}^*$ be the attack strategy that $\hat{g}^*(u', v') = u'$. When $\hat{g} = \hat{g}^*$, the equality achieves, and thus $\hat{g}^* \in \arg\sup_{\hat{g}} \mathbb{P}(\hat{g}(\theta') \in [g(\theta) - \epsilon, g(\theta) + \epsilon])$.

For the distortion, we can easily get that $\Delta = \frac{s}{2}$. Corollary 2 says that $\Delta > \left( \lceil \frac{s}{2\epsilon} \rceil - 1 \right) \cdot \epsilon$. The gap between these two goes to zero when $\epsilon \to 0$. $\qquad\square$

