# OpenReview forum: "Distributional Privacy for Data Sharing"
_NeurIPS.cc/2022/Workshop/SyntheticData4ML — Neurips 2022 SyntheticData4ML_

### Official Review · Reviewer_1dc6 · 2022-10-11
**Authors present a new privacy framework for defining, analyzing, and protecting distributional privacy concerns while data sharing.**

**Rating:** 6
**Confidence:** 5

**Review:**

Overall a well-written paper that delves into the very important topic of privacy in the context of data sharing.
Few comments regarding the manuscript that should definitely be worked on/ addressed,

1. It was not clear what sort of datasets the proposed framework is best suited for? Is it meant only for the cross-sectional dataset as shown in the results section?

2. How do the authors intend to extend this for sequential/ transactional datasets

3. The data release mechanism was demonstrated in the context of a single factor (by adding some stochastic noise). What if there is a privacy risk at a multi-variate level? i.e. Often adversaries will have access to multiple Quasi-identifiers.

4. Shifting the mean does not seem like a very strong method of providing privacy as it still retains the shape of the distribution which in itself can provide lots of information.

5. On page 6, line 226, the authors mention that they resort to surrogate metrics as the data holder only has access to one dataset. This raises a few confusions, (a). Who is the data holder here? (b). in the surrogate metric equations, how come both the original and released datasets are used for evaluation? (contrary to point 5). (c). In the surrogate privacy metric equation, the authors mention a smaller difference between the means of the target variable (between original and private) means strong privacy. I would argue it's the other way. If the difference in the mean is low that would mean, utility is high but low privacy as the adversary can easily obtain the original secrets.

6. The authors should have also looked at some traditional methods both for privacy (i.e. membership attacks, re-identification attacks, etc.) and utility (such as comparison of correlation/ ML utility comparison/ univariate distribution analysis, etc.) to substantiate the claims better in an empirical manner.

---

### Official Review · Reviewer_4hXJ · 2022-10-17
**Distributional Privacy for Data Sharing**

**Rating:** 6
**Confidence:** 3

**Review:**

The paper describes different privacy-preserving algorithms and privacy concerns for the underlying distribution of data in the context of data sharing. Below are some points that need to be addressed in the paper:
1) The paper should cover evaluations of privacy in the skewed distribution scenario of the data.
2) Author should describe more about attacks (such as inference attacks like membership and attribute attacks) and how distributional privacy defends against them.
3) The dataset structure is not very clear in the paper. it will be good to cover the assumption and limitations around the dataset types such as time series data or tabular data or cross-sectional data and their respective evaluation criteria.
4) Evaluation of utility is missing. If data is perturbated based on univariate, then what’s privacy impact on the multivariate level?
5) Adding more benchmarking scenarios will help with the generalization of the proposed approach
6) Is there any alternative way to describe metrics instead of surrogate metrics? Will the surrogate metrics equation define the privacy evaluation between released data and original data held by data holder?

---

### Meta-Review · Area_Chair_ckYu · 2022-10-20

**Recommendation:** Accept